# Multi-Criteria Decision Analysis for Assessing Social Acceptance of Strategies to Reduce Antimicrobial Use in the French Dairy Industry

**DOI:** 10.3390/antibiotics12010008

**Published:** 2022-12-21

**Authors:** Diego Manriquez, Maiara Costa, Ahmed Ferchiou, Didier Raboisson, Guillaume Lhermie

**Affiliations:** 1CIRAD, ASTRE, Université de Toulouse, ENVT, 31300 Toulouse, France; 2Department of Animal Sciences, Colorado State University, Fort Collins, CO 80523, USA; 3School of Veterinary Medicine, University of Calgary, Calgary, AB T2N 4Z6, Canada

**Keywords:** antimicrobial resistance, livestock, decision analysis, public health

## Abstract

To respond to the antimicrobial resistance (AMR) threat, public health entities implement policies aiming to reduce antimicrobial use (AMU) in livestock systems, in which policy success and sustainability might be subject to the social acceptability of the novel regulatory environment. Therefore, consistent methods that gather and synthesize preferences of stakeholder groups are needed during the policy design. The objective of this study was to present a methodology for evaluating the acceptability of potential strategies to reduce AMU using multi-criteria decision analysis (MCDA) using French dairy industry as a model. Preference-ranking organization methods for enrichment evaluations were applied to rank stakeholders’ acceptance of four different potential AMU reduction strategies: 1. Baseline AMU regulations in France; 2. Total interdiction of AMU; 3. Interdiction of prophylaxis and metaphylaxis AMU; and 4. Subsidies to reduce AMU by 25%. A total of 15 stakeholders (consumers, *n* = 10; farmers, *n* = 2; public health representatives, *n* = 3) representing the French dairy sector and public health administration participated in the acceptance weighting of the strategies in relation with their impact on environmental, economic, social, and political criteria. We established a MCDA methodology and result-interpretation approach that can assist in prioritizing alternatives to cope with AMR in the French dairy industry or in other livestock systems. Our MCDA framework showed that consumers and public health representatives preferred alternatives that consider the restriction of AMU, whereas farmers preferred to maintain baseline policy.

## 1. Introduction

Antimicrobial resistance (AMR) is one of the most serious threats for public health [1]. By the year 2050, 10 million deaths could occur worldwide if the current trend of inappropriate and excessive use of antimicrobial agents continues [2]. Antimicrobial resistance occurs when bacteria, virus, fungi, and parasites change over time and no longer respond to medicines, making infections harder to treat and increasing the risks of disease spread, illness severity, and death [3,4]. Two main factors contributing to AMR have been identified. First, the use of antimicrobials that inhibit susceptible bacteria and allow resistant bacteria to survive, and second, the activation of dormant resistance genes due to antibiotic pressure [3,5,6]. Therefore, ecological niches with continuous exposure to antimicrobials can be significant drivers for AMR [7]. In this sense, animal farming is a critical component in the emergence and transmission of AMR organisms [8,9,10].

In livestock systems, antimicrobials play a significant role in treating and preventing disease that cause animal suffering and decrease in productivity [9,11,12]. In dairy farming, antimicrobials are used for the control and treatment of subclinical and clinical mastitis, retained placenta, metritis, lameness, and respiratory disorders [6,13,14]. Despite these benefits, practices around antimicrobial use (AMU) by farmers and veterinarians and the development of AMR due to over-using, inappropriate dosage, incorrect treatment duration, or drug choice are of great concern for the dairy industry and public health institutions [14].

There is consensus that the AMR threat must be approached from a One Health perspective [7,15]. In this sense, the role of stakeholders implicated in the AMR development, as well as the communities at risk and policy makers, must be clearly identified. In France, efforts to control AMU include a national program combining measures to identify and prevent infectious livestock disease, controlled prescriptions, farmer and veterinarian education, and enforcement of current national and international regulations of AMU [16]. Concerning French dairy farming, mastitis is the most commonly reason reported for AMU [14] and the use of critically important antimicrobials (third- and fourth-generation cephalosporins) is allowed only for sick cows and banned for prophylaxis [17,18]. 

Farmers might face challenges to adopt policy against AMR when regulations do not consider their interests and needs. These challenges range from a lack of cost-benefit analyses of using antimicrobials or not, to farmers’ experience detecting and treating infectious diseases, and to welfare considerations of not using antimicrobials [19,20]. On the other hand, consumer perceptions on AMR exert social pressure to improve animal farming practices towards the reduction in AMU, which drives the demand for food produced respecting the environment, animals, and public health. Thus, finding a balance between the main concerns of all stakeholders is necessary to maximize the benefits to farmers, to reduce risks to animal and human health, and to find a rational AMU in dairy farming and in other livestock operations. In this sense, the preference ranking organization methods for enrichment evaluation (PROMETHEE) approach, nested in the multiple-criteria decision analysis (MCDA) spectrum, is a useful tool to assist in the decision-making process on AMU regulations because it can include views from all potential stakeholders, rank strategies according to several criteria and preferences, and prioritize the acceptability of current and novel policy [21]. In consequence, our objective was to develop a MCDA framework for assessing the social acceptability of potential strategies for reducing AMU in animal agriculture, using the French dairy sector as model. 

## 2. Results

The preference weights of different criteria that gauged the impact of the potential AMU policy scenarios were estimated based on stakeholders’ interviews with consumers, farmers, and public health representative groups. After the interview invitations were distributed to potential participants, a total of 10 responders from the consumers group (residents of Toulouse, France), 2 from the farmers group (from Occitanie region, France), and 3 from the public health representatives group (French Ministry of Agriculture, OIE, trade union for the pharmaceutical industry and veterinary diagnostic (SIMV, France)) responded to the interview. A model of the interview is presented in Appendix A.

The acceptability rankings of strategies to reduce AMU were performed within each stakeholder group using the overall performance score (phi) of each strategy, determined by criteria weighing. The consumers’ group score ranged from −0.09 to 0.15 (Table 1). In this group, the first ranked strategy was the total interdiction of AMU with a score of 0.15. The interdiction of prophylaxis and metaphylaxis of AMU was ranked second with a score of 0.01, followed by the strategy subsidizing the reduction AMU by 25% with a score of −0.08 and the baseline strategy with a score of −0.09. In the farmers group, the scores ranged from −0.05 to 0.10 (Table 1). The first ranked was the baseline strategy with a score of 0.10 followed by the total AMU interdiction with a score of −0.02 and the subsidies to reduce AMU by 25% with a score of −0.03. The least accepted strategy was the interdiction of prophylaxis and metaphylaxis AMU with a score of −0.05. On the other hand, the scores of the public health representatives ranged from −0.06 to 0.12. In this group, the total interdiction of AMU was in the first place of acceptation with a score of 0.12, followed by the interdiction of preventive and metaphylactic AMU with a score of −0.01, the baseline strategy, and contrary to the other stakeholder groups, the least accepted strategy was the subsidies to reduce AMU by 25% with a score of −0.06.

Figure 1 shows the multi-criteria problem graphically, considering the acceptability of each stakeholder and the strategies around a decision axis (red line in Figure 1). The exact positions are defined by the stakeholder’s weights and performance of strategies considering each criterion. The closer a stakeholder was to the decision axis, the greater is their agreement with the first ranked strategy. For instance, in Figure 1A, the stakeholder 10 from the consumer group agreed more strongly with the first ranked solution of this group (total interdiction of AMU). On the other hand, stakeholders 3 and 8 were the most distant to this strategy and they disagreed, mostly, in terms of rankings. In the consumers group, all stakeholders were located on the right side of the Y-axis, meaning that their overall preferences were not discordant at all. Stakeholders with longer axis have strong decision choice power [22]. The strategy that considered total interdiction of AMU (AMU interdiction in Figure 1A) was the closest to the decision axis, meaning that it was the preferred strategy by the consumers group. Interdiction of prophylaxis and metaphylaxis AMU was on the right side, since it was placed second in the ranking (Table 1). On the contrary, the least preferred strategies, baseline and subsidies to reduce AMU, were located on the left side of the Y-axis. 

A similar interpretation can be employed for preference ranking of stakeholders from the farmer (Figure 1B) and the public health representative (Figure 1C) groups. In the farmers’ group, the baseline strategy was closer to the decision axis; therefore, it was the most preferred strategy. The participating farmers were not discordant with the overall acceptability of the baseline strategy, since they were located on the right of the Y-axis. Nonetheless, they differed in in terms of preferences and criteria rankings due to the separation between the stakeholders’ axis (Figure 1B). On the other hand, farmers did not prefer the remaining three strategies at all. Finally, in the public health group, total interdiction of AMU was the closest to the decision axis (Figure 1C), although the first-ranked strategy was not as close to the decision axis as the strategies preferred by consumers and farmers groups (Figure 1A,B). This might be due to more dissimilar criteria ranking among public health stakeholders. For instance, in Figure 1C, stakeholder 3 is pointing to the left side of the Y-axis because the baseline strategy was their first-ranked strategy. Stakeholder 1 is the closest to the decision axis, meaning that they have the best agreement with the first ranked strategies. Stakeholders 2 and 3 were very distant in their rankings.

To assess group acceptability, an overall decision map was developed considering the preferences of the three groups of stakeholders. Therefore, in Figure 2, consumers, farmers, and public health representatives were aggregated in an individual axis by averaging their weights for each criterion. For the consumer and public health groups, the AMU interdiction (Figure 2) was the closest to the decision axis, since this was the preferred strategy for both groups. On the other hand, the farmers’ group had baseline strategy closest their axis. The two strategies with the least preference for all stakeholders were on the left side of the Y-axis (Figure 2).

The action profile tool on the VP software was used to represent graphically the impact of each strategy on each criterion considering all stakeholder groups (Figure 3). According to the interviewees’ preferences, the baseline strategy (Figure 3A) performs well in the production costs and in the social and political criteria; however, it performs poorly in most economic and environmental criteria. The total interdiction of AMU (Figure 3B) performs well in almost all economic and environmental criteria; however, it performs poorly in production cost and in social and regulatory framework criteria. The subsidies to reduce AMU by 25% (Figure 3C) perform badly in almost all economic, environmental, and policy investment criteria; nonetheless, they perform more efficiently on the production cost, regulatory framework, and social criteria. Finally, the interdiction of prophylaxis and metaphylaxis AMU (Figure 3D) strategy performs well in farmer’s revenues and reasonably well in the attributable fraction, mortality rate, and policy investments. Conversely, it is not efficient in the other economic criteria and in the regulatory framework.

To evaluate whether changes in criteria weighing could have an impact on the analysis, a sensitivity analysis was performed using the stability intervals window of the VP software. Table 2 shows the weight stability intervals for each criterion. Weights outlying this intervals could affect stakeholders’ rankings of the preferred strategies. The sensitivity analysis showed that the stakeholders can change the weights due to policy investments from 7.53 to 100 without affecting the stakeholder’s ranking strategies. However, any variations in the weight of the production cost, for example, beyond the range of 12.92 to 15.22 will result in a change in the ranking, indicating that results are more sensitive to this criterion. Due to short stability intervals, other sensitive criteria were farmers’ revenues, culling rate, regulatory framework, and the attributable fraction (Table 2).

## 3. Discussion

We aimed to present a MCDA framework including implementation and result interpretation that could be used to assess the social acceptability of a novel policy environment against AMR in the French dairy industry. Although we performed comparisons between strategies to reduce AMU to exemplify our MCDA approach, we did not intent to test the viability of these strategies in real settings. Moreover, we acknowledge the limitations of the strategy comparison due to the low response rate. Therefore, we encourage readers to use the information presented here as a framework to perform MCDA around the AMU policy development. 

The number of AMR microorganisms is rising at high levels in all parts of the world [1,3,21]. Therefore, fighting AMR requires a multidisciplinary approach [23]. In order to succeed, concatenated efforts involving the needs and visions of diverse members of the society require methodologies and metrics that estimate the appropriateness of interventions aiming to reduce AMU [23]. In this sense, MCDA can be a key component for addressing AMR, since its conclusions assist in finding an informed balance for decision makers, considering the interests and ethical concerns of diverse stakeholders. Moreover, MCDA not limited to the consumers, farmers, and public health groups that we considered to develop our approach. 

In the dairy industry, antimicrobials are critical for the control and treatment of mastitis, which produces significant economic loses at the farm level worldwide, and other disorders that cause economic losses and animal suffering [24]. In France, mastitis was considered responsible for a third of the economic impact related to health disorders in dairy cows [25]. Moreover, every dairy cow receives on average the equivalent of 1.58 antimicrobial treatments for mastitis per year, which represents 70% of all the AMU in dairy cows [26]. To respond to AMR, a more restrictive usage of antimicrobials by the livestock industry has been suggested to prevent outbreaks of multi-resistant bacteria and avoid the spread of resistance genes to other hosts and communities [3,5]. 

When public policies are formulated, the complexity of the disease, the diversity of farm environments, a farm’s microbial ecology, and the idiosyncrasy of farmers should be considered. Nonetheless, it is common to observe that AMU policy only considers one side of the problem, such as total restriction, which might affect other management practices and not be sustainable in the long-term. In this study, we proposed to participants four potential strategies that have not been fully implemented. However, these strategies synthesize potential scenarios of AMU policy and allowed us to develop our MCDA framework. Regarding these strategies, no country has yet established a regulation that totally bans AMU in animal agriculture. Nonetheless, special productive settings, such as voluntary organic dairy production, have been already implemented posing new challenges to dairy farmers regarding health and reproduction management [27]. Additionally, the European Union has banned AMU as growth promoters [28,29] and has fostered the antimicrobial-free production. In France, some regulations supervise the use of third- and fourth-generation cephalosporine and have implemented a plan to reduce AMU by 25% during a 5-year period [16]. Concerning the implementation of direct subsidies for farmers to encourage AMU reduction, experiences are scarce, and the long-term effects have not been fully explored [20]. However, few programs have been implemented. These programs usually come from the private sector, where manufacturers pay premiums for animal products raised without antibiotics [30]. On the other hand, in the public sector, tax systems have been preferred to reduce antimicrobial sales [23,31]. 

The introduction of new regulations for controlling AMU affects production costs and product prices. For instance, in the USA, it has been estimated that the interdiction of prophylaxis AMU would cause loses of $1.8 billion USD for the beef industry [32] and, in the dairy industry, a reduction in AMU may lead to increased morbidity and/or mortality and reduce the production output for the milk supply chain [33]. Moreover, interventions that consider antimicrobial prohibition or abrupt reduction in AMU must have an ethical approach because of the impact on animal welfare and economic consequences in veterinary care, agriculture, and relevant bio-industries. For farmers, these restrictions may threaten the quality and quantity of animal food products. Additionally, low-income countries may face greater challenges to meet production goals or compete with larger markets [29]. In this sense, the assessment of AMU sustainability is necessary to advise policymakers about the possible impact on stakeholders’ view and acceptability [20]. 

To perform sustainability policy assessments, decision support tools are frequently used, in which the most common are risk analysis, cost-benefit analysis, system dynamics, and MCDA. In the context of AMR, risk analyses are useful before the implementation of policies for assessing and managing human and animal health risks associated with the development of resistance, including appropriate communication measures. Nonetheless, the implementation of risk analysis is costly and time consuming because it requires several steps to be completed, including hazard identification, risk assessment, risk management, and risk communication [34]. In this study, we used MCDA because it provides a structured and systematic process for identifying gaps in scientific knowledge related to important decision issues, which can be used to outline research priorities in public health. In addition, MCDA can be adapted and potentially used with real-time decision-making methods, and it supports individuals or groups of decision makers to classify, select, and compare different alternatives for solving a problem [21]. Another advantage of MCDA is that multiple comparisons are viable when there are competing and multiple evaluation criteria. Moreover, the MCDA methods can include quantitative and qualitative data in the analysis and, contrary to a cost-benefit analysis, they do not assign a monetary value, which is extremely difficult to estimate for environmental and social impacts. Additionally, MCDA allows one to include weights or perspectives from all parties involved in AMR [35].

Despite the strategies designed by the French administration, there has been an increase in exposure to antimicrobials in dairy cows between 2017 and 2018 [16]. Therefore, the assessment of new strategies for reducing AMU are continuously needed. In this sense, MCDA and PROMETHEE provide a useful and simple procedure for assessing the acceptability of innovative strategies against AMR. To the authors’ knowledge, this is the first report that applies this framework to analyze a potential scenario of AMR regulations applicable to the dairy industry. 

The PROMETHEE procedure is based on the pair-wise comparison of strategies. In this case, the deviation between the evaluations of two alternatives on one particular criterion is considered. Small deviations indicate weak preference or no preference, as the stakeholders will consider this deviation small or negligible. For larger deviations, higher preference levels are expected. With PROMETHEE, preference levels are measured on the degree of preference flow (phi) between −1 to 1, where −1 means no preference and 1 means total preference [36]. Additionally, PROMETHEE relies on a preference function indicating the degree of preference from one alternative over the other [37]. In this study, the entire decision analysis was carried out in VP software, which greatly facilitates the performance assessment of strategies on each criterion [21].

From our MCDA approach, we determined that the first-ranked alternative for consumer and public health representative groups was related to the total interdiction of AMU (Table 1). This agrees with other studies suggesting that consumers prefer to buy antibiotic-free products when only the AMR issue is evaluated [38]. Therefore, it is plausible to suggest that consumers will support antimicrobial prohibition for livestock systems. On the other hand, public health administration is expected to improve the education efforts about AMU and AMR, to strengthen surveillance and research, optimize AMU in human and animal health, and develop sustainable investment policies for new medicines, diagnostic tools, and other interventions [39]. Thus, MCDA may be useful to prioritize educational topics for specific stakeholders.

Other strategies presented in this study had lower scores and, therefore, were less preferred by consumers and public health representatives (Table 1). In the individual rankings, this issue was evident because only one stakeholder from these two groups did not select AMU interdiction as the first-ranked choice. The only differences found between the consumers and public health groups were between the third and fourth position in the ranked strategies (Table 1). Conversely, farmers performed a ranking where the baseline strategy was preferred. This might be explained because farmers may see their managements and costs and benefits at risk [20]. Overall, most stakeholder weighing was allocated to the economic dimension with an average of 43.84 points for the consumers, 37.33 for the farmers, and 59.33 for the public health representatives. This shows that this dimension is of great importance for all stakeholder groups. However, larger number of participants will be needed to make robust comparisons between the potential AMU reduction strategies.

The methodology presented in this study is highly flexible in incorporating more stakeholders and blocking stakeholders by demographic features, relevant criteria, and other productive contexts to take better decisions regarding AMU and AMR. Nonetheless, future research using our MCDA approach will require a larger number of participants, considerations of demographic bias, and a comparable number of stakeholders in each group to assure the external validity of conclusions made from MCDA in AMU policy design.

## 4. Materials and Methods

### 4.1. Study Design

This study was conducted from January to July 2020 with the general objective of creating a MCDA framework for assessing potential scenarios of AMU policy, allowing one to rank strategy preference from consumers, farmers, and public health representatives in the French dairy sector. Within the MCDA methodology, the PROMETHEE approach was used. The study started planning a general framework, which is necessary for carrying out the PROMETHEE technique.

To begin this framework, we selected criteria which can be assessed under environmental, economic, social, and political dimensions of AMR and AMU. The environment dimension had two criteria: AMU assessed by the animal level of exposure to antimicrobials (ALEA) and AMR assessed by the fraction of AMR human infections attributable to livestock. The economic dimension had three criteria: production costs, farmer’s revenues, and product prices (meat and milk). The social dimension was assessed using animal welfare metrics, including herd culling rate and mortality. Finally, the political dimension criteria consisted of the number of policies and investments to fight AMR. Following this, we identified stakeholders in AMU policies comprising three groups of consumers of dairy products, dairy farmers, and public health representatives in the region of Occitanie, France. The proposed MCDA framework for criteria and stakeholder selection is depicted in Figure 4. After criteria and stakeholder identification, stakeholders individually weighed the impact that four potential strategies to reduce AMU will have on each criterion. The strategies were: 1. Baseline AMU regulations in France; 2. Total interdiction of antimicrobials; 3. Interdiction of prophylaxis and metaphylaxis AMU; and 4. Subsidies to reduce AMU by 25%. Following this step, group criteria weighing was analyzed using PROMETHEE, in which strategy preference was compared within and between stakeholder groups. 

### 4.2. Development and Assessment of Criteria

We translated the issues related to AMR and AMU into four dimensions with measurable assessment criteria within them. In the environmental dimension, we used the ALEA indicator as an estimation of antimicrobial sales and animal exposure to antimicrobials [40]. The ALEA indicator is used by French authorities to report and monitor yearly antimicrobial sales [41]. This criterion estimates the percentage of animals treated with antimicrobials out of a total animal population. It is calculated as follows: $$ALEA=\frac{Live\;weight\;treated}{\left(Total\;number\;of\;animals*Weight\;of\;adults\;animals\;or\;at\;slauther\right)}$$

Table A1 shows criteria values for criteria for each strategy scenario. Current estimations performed by ANSES (2018) [41] set ALEA at 0.27 for bovines. Therefore, this value was attributed to ALEA in the baseline strategy for AM reduction. Additionally, in the environmental dimension, we used the estimated attributable fraction of AMR human infections associated with animal agriculture. The CDC (2013) [42] estimated that one out of five AMR bacterial infections are linked to food or animals, but an accurate fraction of AMR human infections attributable to dairy cattle is unknown. Various experts estimated that the overall contribution was about 4% in 2000 [43]. Nonetheless, due to the complexity of the phenomenon and the difficulty in evaluating it, this appears largely underestimated [44]. For this study, the value of 4% for the attributable fraction was assigned to the baseline strategy. In the economic dimension, farmers’ costs and revenues and product prices were selected as measurable criteria. The costs and revenues were measured by the milk production costs, the revenues from the milk price, and the average price of culled cows. The product price is defined by the selling price of a liter of milk. The farmers’ costs were estimated from the current expenses, depreciation, and additional expenses. The costs were set at 494 euros/1000 liters of milk [45] and the revenues at 0.78 cents/liter of milk [46]. Additionally, the average price of culled cows was set at 2.4 euros/kg net [47]. These values were assigned to the baseline strategy. In the social dimension, animal welfare was assigned as a measurable criterion due to the impact that reducing AMU might have on animal wellbeing and reflected by changes in mortality or live culling and morbidity. In France, the estimated annual mortality and culling rates are 3.8% and 21.3%, respectively [48]. These rates were used in the baseline strategy. Finally, the political dimension was evaluated using the regulatory framework concerning AMR and policy investments. These criteria were measured in a semi-quantitative way (null, low, moderate, high, very high). The idea was to show stakeholders that greater regulations and investments related to strategies for reducing AMU are also associated with increased costs to the population, in the form of taxes. 

### 4.3. Definitions of Strategies against AMR

The four strategies used in this study were created with the purpose of developing a MCDA approach. The baseline strategy corresponds to the current situation of antimicrobial use in French dairy farms. The current French strategy against AMR has led to a 37% reduction in the veterinary AMU between 2012 and 2016 [16,49]. This strategy assumes that AMU will remain being critical for dairy cattle, especially for mastitis control and treatment of pneumonia in calves [19,50]. The total interdiction of antimicrobials assumes an AMU ban at all production stages in dairy farms. We assumed that no substitution treatment or alternatives are implemented. For the interdiction of prophylaxis and metaphylaxis of AMU, it was assumed that these uses correspond to 35% of the total AMU in French dairy farms. Subsidies to reduce AMU by 25% consist in encouraging producers to adopt desirable practices to reduce AMU and receive monetary compensation. Here, the hypothesis was that farmers will manage to reduce the use of antimicrobials by 25% and they will receive subsidies.

### 4.4. PROMETHEE Implementation

The general framework depicted in Figure 1 was applied using the PROMETHEE method as developed by Behzadian et al. (2010) [51] and Aenishaenslin et al. (2013) [52]. To use PROMETHEE and assess societal acceptance of strategies for reducing AMU in the French dairy sector, we carried out the following steps:

#### 4.4.1. Problem Definition and Identification of Stakeholders

To address the general problem of identifying, evaluating, and ranking different strategies to decrease AMR in dairy farms in France, three groups of stakeholders (consumers, farmers, and public health representatives) were invited to participate in the weighing-criteria process. We aimed a convenience sample size of 10 responders from each stakeholder group. A stakeholder was defined as a person representing an organization or a group with direct responsibilities or with specific interests in AMU in dairy cattle [52].

#### 4.4.2. Identification of Key Decision Issues and Definition of Criteria

Criteria were identified to evaluate the effectiveness of AMU strategies. These criteria were nested into four dimensions: environmental, economic, social, and political. Defined criteria had scaled values derived from our experience and literature to calibrate the PROMETHEE model that might determine decision making of stakeholders under each strategy (Table 3).

#### 4.4.3. Weighing Criteria and Criteria Group Ranking

After the criteria and the strategies to be evaluated have been defined, a total of 10 residents of Toulouse as part of the consumer group, 10 dairy farmers from the Occitanie region, and 10 public health representatives from the French Ministry of Agriculture, OIE, the National Veterinary School of Toulouse, the French National Agency for Food, and the Environmental and Occupational Health Safety Agency were invited to participate in the interviews using the interview model shown in Appendix A. Following this, stakeholders who responded to our invitation participated in the criteria weighing. Here, stakeholders were asked to distribute a total of 100 points among the criteria in order of importance, considering how the strategies will affect the criteria as shown in Table 3. The most important criteria for them should receive more points.

#### 4.4.4. MCDA

The Visual PROMETHEE software (VP; ULB, Brussels, Belgium) was used to perform MCDA and obtain comparative rankings of a set of alternatives. Overall numerical scores assigned by the participants were entered for each criterion on VP. Following this, results were visualized using a visual model performed by the GAIA (Graphical Analysis for Interactive Aid) tool provided in VP. Group rankings of strategies yielding the best to worst alternatives were performed using the individual stakeholder’s values expressed via criteria weighing.

#### 4.4.5. Interpretation of Results

Each stakeholder was asked to weight each criterion after observing the impact of each strategy on criteria (Table 3). Preference flow values were calculated using the VP software. These values are consolidated results of the preferences of each strategy and allow pairwise comparisons between strategies. The positive preference flow (phi+) allows one to measure the extent of agreement of one stakeholder with a strategy versus the other strategies and it represents a value of the strength of acceptability for one strategy. On the other hand, the negative preference flow (phi−) measures how much other strategies are preferred versus a particular strategy. Thus, phi− is a global measure of the weakness of a strategy in terms of acceptability. Finally, the two parameters are combined and result in the net flow score (phi), which provides the overall performance score for each strategy. Values greater than 0 are more preferred, conversely, values lower than 0 are less preferred.

To model the way the stakeholder perceives the criterion measurement scale, the PROMETHEE procedure requires associating a preference function with each criterion. The V-shape preference function was used to analyze quantitative criteria [21]. This preference function is a special case of the linear preference function where the Q indifference threshold is equal to 0. This method is efficient for quantitative criteria when even small deviations are accounted for [37]. 

#### 4.4.6. Sensitivity Analysis

A sensitivity analysis was performed using the Visual Stability Intervals tool in VP for assessing the impact of a stakeholder’s weighing preferences on their individual and group rankings. This analysis gives indications of the robustness of the results and can be generated for each stakeholder for all criteria.

## 5. Conclusions

Our MCDA framework offers a methodology to assess potential strategies to reduce AMR development in the French dairy industry. In our approach, we observed that the most accepted strategy for consumers and public health representatives was total interdiction of AMU, whereas farmers opted to maintain the baseline strategy. Although the strategies considered in this study might not be plausible in the current dairy industry and due to the low number of responders, this study provides a MCDA framework that can be applied with an ample range of strategies for assessing their acceptability by relevant stakeholders in the food-animal supply.

## Figures and Tables

**Figure 1 antibiotics-12-00008-f001:**
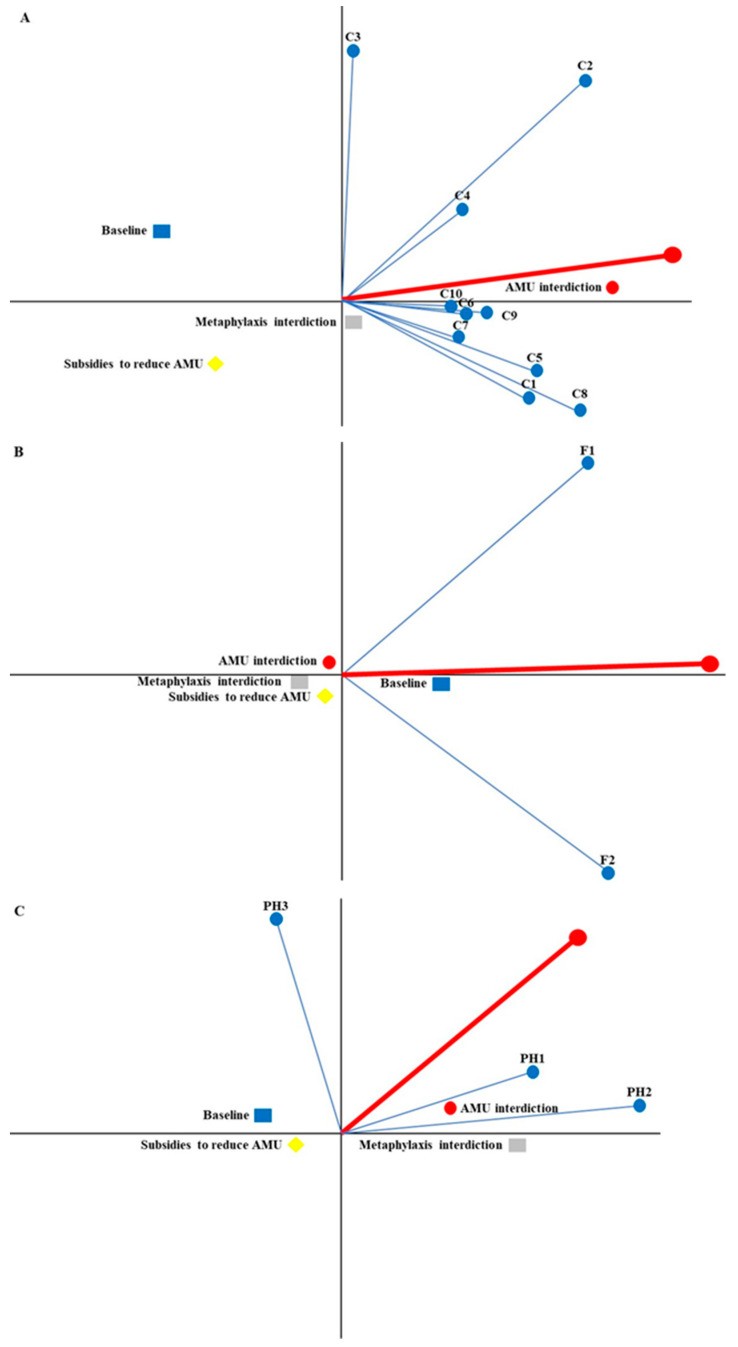
Decision map (GAIA plane) of consumers (**A**), farmers (**B**), and public health (**C**) decision makers weighted preferences. (Delta = 96.6%; 96.6% of the information is conserved in the two-dimensional representation of this map).

**Figure 2 antibiotics-12-00008-f002:**
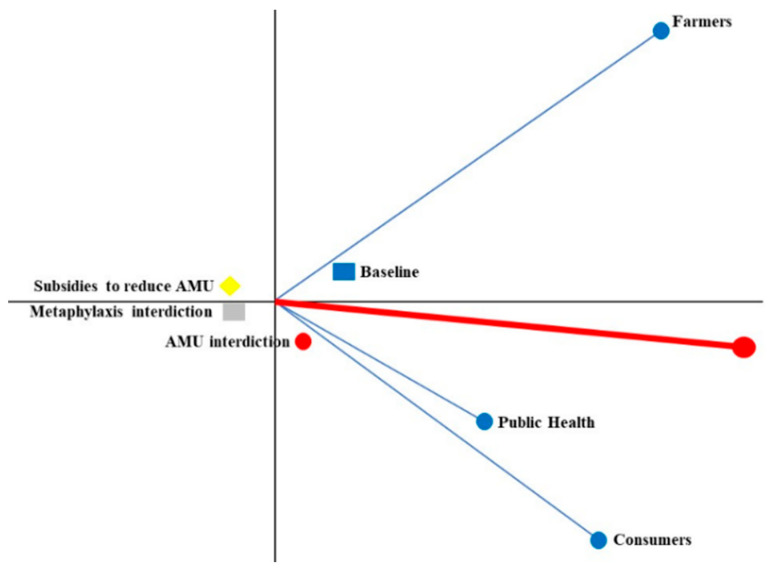
Aggregated decision map (GAIA plane) of decision makers considered in this study. (Delta = 96.6%; 96.6% of the information is conserved in the two-dimensional representation of this map).

**Figure 3 antibiotics-12-00008-f003:**
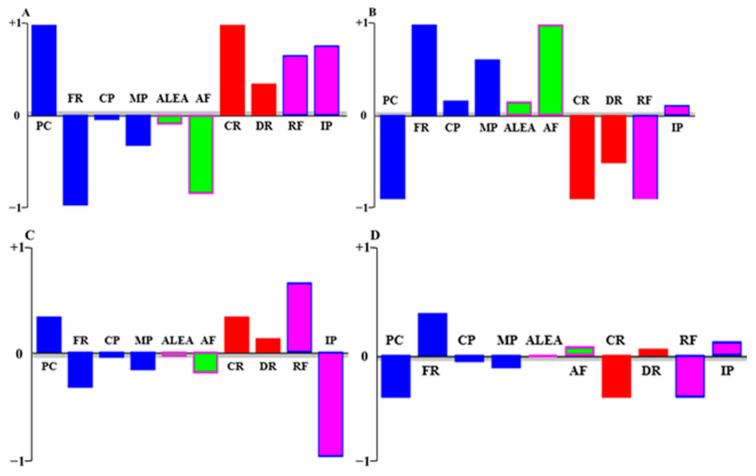
Performance profile of strategies to reduce antimicrobial use under each criterion. The strategies included were baseline strategy of antimicrobial use (**A**), total interdiction of antimicrobial use (**B**), subsides to reduce antimicrobial use by 25% (**C**), and interdiction of preventive and metaphylactic antimicrobial use (**D**). Abbreviations. Economic dimension (blue): Production costs (PC), farmers’ revenues (FR), culled cow price (CP), milk price (MP). Environmental dimension (green): animal exposure level to antimicrobials (ALEA), attributable factor of antimicrobial-resistant human infection to livestock (AF). Social dimension (red): culling rate (CR), death rate (DR). Political dimension (pink): regulatory framework (RF), investment policies (IP).

**Figure 4 antibiotics-12-00008-f004:**
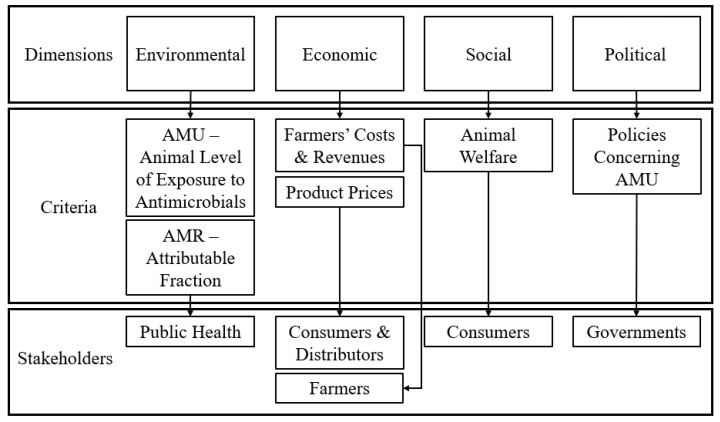
Multi-criteria decision analysis flowchart used to determine criteria and stakeholder groups involved in AMU in French dairy industry. AMU: antimicrobial use. AMR: antimicrobial resistance.

**Table 1 antibiotics-12-00008-t001:** Acceptability ranking scores of strategies to reduce antimicrobial use in French dairy farms by stakeholder group.

Stakeholders	Weighted Ranking	Strategy	Phi	Phi+	Phi−
Consumers	1	AMU interdiction	0.23	0.58	0.22
	2	Preventive AMU interdiction	0.007	0.24	0.25
	3	Subsides to reduce AMU	−0.10	0.27	0.37
	4	Baseline strategy	−0.19	0.19	0.32
Farmers	1	Baseline strategy	0.1	0.36	0.26
	2	AMU interdiction	−0.02	0.34	0.36
	3	Subsides to reduce AMU	−0.03	0.29	0.32
	4	Preventive AMU interdiction	−0.05	0.28	0.33
Public health representatives	1	AMU interdiction	0.12	0.45	0.33
	2	Preventive AMU interdiction	−0.004	0.25	0.26
	3	Baseline strategy	−0.03	0.34	0.35
	4	Subsides to reduce AMU	−0.09	0.21	0.30

Phi: overall preference flow; Phi+: positive preference flow; Phi−: negative preference flow.

**Table 2 antibiotics-12-00008-t002:** Sensitivity analysis of stakeholder weighing.

Criteria	Weight Stability Interval		
	Minimum	Maximum	Difference
Regulatory framework	9.28	11.12	1.84
Farmer’s revenues	15.84	18.1	2.26
Production cost	12.92	15.22	2.3
Culling rate	5.75	8.24	2.49
Attributable fraction ^2^	2.99	8.84	5.85
Product price	1.31	12.86	11.55
ALEA ^1^	0	11.99	11.99
Mortality rate	5.8	17.94	12.14
Price culled cow	0	12.54	12.54
Policies investments	7.53	100	92.47

^1^ ALEA: animal level of exposure to antimicrobials; ^2^ Fraction of antimicrobial-resistant human infections attributed to animal agriculture origin.

**Table 3 antibiotics-12-00008-t003:** Criteria and scaled measures used in the PROMETHEE models under strategies (STRA) for reducing antimicrobial use in French dairy farms.

	Criteria	STRA01	STRA02	STRA03	STRA04
Environmental	ALEA ^1^	0.27	0	0.17	0.20
	Attributable fraction ^2^ (%)	0.04	0	0.026	0.03
Economic	Production costs (€/1000 L)	494	684	667	617.5
	Farmers’ revenues (€/1000 L)	334	473	451	417.5
	Culled cow price (€/Kg)	2.4	2.64	2.4	2.4
	Product price (€/L)	0.78	1.85	1.05	0.96
Social	Mortality rate (%)	3.8	4.8	4.1	4.04
	Culling rate (%)	21.3	50.5	31.5	28.6
Political	Regulatory framework	Moderate	Very high	High	Moderate
	Investment Policies	High	High	Moderate	Very high

^1^ ALEA: animal level of exposure to antimicrobials; ^2^ Fraction of antimicrobial-resistant human infections attributed to animal agriculture origin; STRA01: baseline current strategy of antimicrobial use in France; STRA02: the total antimicrobial interdiction; STRA03: interdiction of antimicrobial use as prophylaxis and metaphylaxis management; STRA04: the implementation of subsidies to reduce antimicrobial use by 25%.

## Data Availability

Datasets are available upon request to the corresponding author.

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
