# Peer review of "Multi-Criteria Decision Analysis for Assessing Social Acceptance of Strategies to Reduce Antimicrobial Use in the French Dairy Industry"

_antibiotics, 2022, doi:10.3390/antibiotics12010008_

Round 1

Reviewer 1 Report (Previous Reviewer 1)

Thank you for responding to the previous comments.  The paper is now much more focused on presentation of the methodology and there are less unsupported statements about inappropriate antibiotic usage.  The paper could be improved by providing the demographic characteristics of the participants, but apparently that information is not available and the focus on introducing the methodology makes it less critical.  I encourage you to better describe the impact of participant selection on your outcomes and expand your recommendations of how to use this methodology to reach robust conclusions.

Author Response

Dear Reviewer,

            We thank you for your time and efforts to review our manuscript and provide great suggestions. We have responded to your last comment below and in the manuscript.

Reviewer:

Thank you for responding to the previous comments.  The paper is now much more focused on presentation of the methodology and there are less unsupported statements about inappropriate antibiotic usage.  The paper could be improved by providing the demographic characteristics of the participants, but apparently, that information is not available and the focus on introducing the methodology makes it less critical.  I encourage you to better describe the impact of participant selection on your outcomes and expand your recommendations on how to use this methodology to reach robust conclusions.

AU: We added ideas about the importance of considering demographic bias when developing MCDA.

Reviewer 2 Report (Previous Reviewer 2)

I have no further comments on this manuscript. 

Author Response

Dear Reviewer,

            We thank you for your time and efforts to review our manuscript and provide great suggestions to improve the quality of this manuscript.

This manuscript is a resubmission of an earlier submission. The following is a list of the peer review reports and author responses from that submission.

Round 1

Reviewer 1 Report

Overall Comments

The researchers present an interesting analysis that is focused primarily on opinions of non-farm associated consumers about ways to reduce AMU on dairy farms.  This type of social science-based research is needed to ensure that actions taken to impact AMU align with societal expectations.  While use of this methodology leads to predictable outcomes (farmers and selected stakeholders have varying opinions about strategies to reduce AMU) the results do add to our knowledge of some of the gaps between farmers and consumers.  However, several improvements and clarifications are needed in this paper.  The authors need to better describe how the participants in their MCDA were selected and who they represent. The external validity and reference population for this study are dependent on the characteristics of the participants so inclusion of demographic information about the participants is vital to understanding how to apply these results.  Additionally, the authors make a few broad and unsupported statements that infer that there is widespread misuse of antibiotics in the French dairy industry and infer that misuse contributes to development of AMR.  I am sure that the authors do not intend to exaggerate AMU and its impact within the dairy industry and urge them to use more restraint and provide evidence-based citations that reflect AMR and AMU in dairy not in other species.  The use of hyperbole tends to reduce credibility. 

Abstract –

L11.  The statement that “misuse of antimicrobials in animal farming….” Is a very broad statement relative to the subject of this study (the dairy industry).  This reviewer would caution against such broad statements as AMU in dairy cattle is relatively low compared to other animal commodities and I am not aware of studies that have documented misuse of AMU in the French dairy industry.  I suggest that you revise this statement to be more balanced.

Introduction – The focus of the study is on the French dairy sector, but much of the introduction is more general to livestock farming and I have some concerns that the authors overemphasize use of antimicrobials  on dairy farms as an important reservoir for emergence of AMR.  For example, on line 43 they state that “antimicrobials play a significant role for improving production efficiency,”  while this may be true for some sectors it is not relevant to dairy cattle as I am not familiar with use of antimicrobials for this purpose within the dairy industry.  I am concerned about these statements because readers may infer that these non-approved usages of antimicrobials occur commonly on dairy farms, and I encourage the authors to revise their introduction be more specific to AMU within the subject of this research (dairy industry) and to use citations that are specific to their subject (dairy farming) rather than broadly applying to other animal commodities that use antimicrobials much more intensively. 

L33.  Change “continuous” to “continues”

L45.  The sentence that begins “In developed countries, between 50-80% of …..” infers that poultry, swine and dairy contribute equally to purchase of antimicrobials.  The next sentence indicates correctly that AMU is commonly used for control and treatment of various bacterial diseases in dairy cattle.  However, NONE of the references cited (13-15 or 6, 16) support that the dairy industry is a significant contributor to overall mass of AMU.  Indeed, country level sales data from the Netherlands (https://cdn.i-pulse.nl/autoriteitdiergeneesmiddelen/userfiles/sda%20jaarrapporten%20ab-gebruik/ab-rapport-2021/figure-1.jpg ) and the US (https://www.fda.gov/animal-veterinary/cvm-updates/fda-releases-annual-summary-report-antimicrobials-sold-or-distributed-2020-use-food-producing  ) both indicate that use on dairy farms is much less than other animal commodities (sales of intramammary products are <1% of total antimicrobial mass sold in the US for use in food producing animals).  Please rephrase to be more balanced and focused on the subject of this research. 

L54.  I am confused by the use of the Poizat et al., 2017 PVM paper to support that animal farming is a significant player in AMR development as the subject of that paper is not AMR development but rather the influence of external drivers on choices of treatments on dairy farms.  This citation is inappropriate and should be replaced or the statement revised.  

Results:  The results should begin with a description of the participants in the groups.  How many were drawn from each of the organizations listed in the MM?  The gender, age, level of education, professions, location of primary residence (rural or urban) etc. should be described.

L74.  I am uncertain who the six people were that performed the method calibration.  Can you please clarify?

L76.  Why were the participants distributed unequally among the stakeholder groups?  What is the rationale for 10 participants? 

L102-116.  This is a nice description of how to interpret this figure.  Thank you.

Discussion – in general, I would like to have the authors discuss in more depth the limitations of this project.  For example, how does the sample size compare to other similar studies using this methodology?  What are the limitations of the results?  How do the demographics of the participants impact the ability to generalize the results?  What are some potential reasons that fewer farmers accepted the invitation to participate as compared to other groups? 

L192-199.  This is important information but focuses on “restrictive” usage of antimicrobials rather than opportunities for farmers to properly use antimicrobials based on selective therapy of mastitis using culture guided treatments for clinical mastitis and selective dry cow therapy at the end of lactation.  In many parts of the world, adoption of selective treatment strategies for mastitis has reduced AMU for mastitis by about 50%.  Some discussion about optimizing antibiotic usage by promotion of selective culture guided therapy should be included. Indeed, a strategy to incentivize selective therapies would have been a good intervention to include in your assessment.

L208.  While no country has fully banned AMU, various functional bans exist in some commodities for some production systems.  For example, in the US, there is ZERO allowed antibiotic usage for animals (meat and dairy) that are raised under organic standards.  There are several published papers in the US that contrast animal health between organic and conventional systems with little differences observed.  Similarly, some vertically integrated poultry producers have ZERO antibiotic policies.  Thus, these market driven policies should be discussed. Please add these voluntary antibiotic bans to your discussion.

L221.  While you cite that there would be $1.8 billion USD loss for the beef industry, this study is about the dairy industry.  Can you please expand your discussion here to focus on the dairy industry?

L278 – this section would be vastly enhanced if we knew if the participants in the groups were comparable on a gender, educational attainment, economic status etc. basis.  Please expand this discussion after providing that information. 

 Materials and Methods –

L321 (section 4.1) – The ALEA is apparently the French metric for sales of AM, but can you specify if the 0.27 estimate is specific for dairy farming or applies across all animal types?  If it is a general estimate then it likely greatly overestimates usage in dairy production.  Again – I am quite concerned that we are mixing apples (overall AMU in AMU intensively raised meat animals– such as pigs) with oranges (AMU in highly regulated dairy cows that require milk discard when antibiotics are used).  Please clarify the population that this ALEA applies to and is drawn from. 

L336 – please provide support that 4% AF applies to dairy production.

L354 (section 4.2) – very useful and well described sections

L377-382 – can you please describe how you selected stakeholders and decision makers?  Was there a criterion relative to how to balance the various groups?  This is an important issue as previous research in the US has shown that stakeholders and consumers vary enormously in their opinions about AMU on dairy farms (see 1.   Wemette M, et al. (2020) New York, State dairy farmers’ perceptions of antibiotic use and resistance: A qualitative interview study. PLoS ONE 15(5): e0232937. https://doi.org/10.1371/journal.pone.0232937   and 2.  Wemette, M., et al., Public perceptions of antibiotic use on dairy farms in the US.  JDS 104:2807-2821 https://doi.org/10.3168/jds.2019-17673).

L390 – please describe why these organizations were chosen, how many people were invited and declined and the characteristics of the participants.    Please consider discussing the impact of who participated in your groups and the impact of your selection criteria on the outcomes of your study.  For example, clearly a greater proportion of consumers who were invited participated as compared to dairy farmers.  How does the difference in participation impact the generalizability of your results?

Reviewer 2 Report

Thank you for the opportunity to review this manuscript. This paper is needed. We need to consider all these scenarios to find the best intervention to reduce AMU and tackle AMR. The paper is well-written and the figures are understandable and explanatory.  However, I’m not sure about the methodology and the large number of assumptions that need to be made in order to draw these results. In addition, I consider that the number of participants per group of stakeholders is extremely low. I would love to see this paper published, but with greater power (sample size) to extrapolate results, or at least to have some validity in the target population (dairy farmers in France). It is important to consider the large number of factors that affect AMU and the unintended consequences of AMU restriction; therefore, it is important to add more producers/authorities/consumers to investigate this in a more powerful study.

Reviewer 3 Report

The manuscript uses a Multi-Criteria Decision Analysis method to evaluate the acceptability of strategies for reducing antimicrobial use among 15 stakeholders and rank the best strategies for participants. Unfortunately, a lot of interesting points including more feasible strategies within the stakeholder was not be taken considered in the structure set, and thus lead to unplausible suggestion in the current dairy industry as the author stated. Thus, we are unable to consider the manuscript further at Antibiotics. We believe that it would be more suitable for another general strategy journal with slightly less stringent requirements, or a more specialized journal where it will be better appreciated by the Analysis framework community.